# Ecological and Construct Validity of a New Technical Level Cuban Dance Field Test

**DOI:** 10.3390/ijerph182413287

**Published:** 2021-12-16

**Authors:** Johnny Padulo, Alin Larion, Olfa Turki, Ionel Melenco, Cristian Popa, Stefano Palermi, Gian Mario Migliaccio, Stefania Mannarini, Alessandro Alberto Rossi

**Affiliations:** 1Department of Biomedical Sciences for Health, Università degli Studi di Milano, 20133 Milan, Italy; 2Faculty of Physical Education and Sport, Ovidius University of Constanta, 900470 Constanta, Romania; alinlarion@yahoo.com (A.L.); ionel_melenco@yahoo.com (I.M.); crispopa2002@yahoo.com (C.P.); 3Higher Institute of Sport and Physical Education of Ksar Said, Tunis 2010, Tunisia; olfa.turki17@gmail.com; 4Research Unit (UR17JS01) Sport Performance, Health & Society, Higher Institute of Sport and Physical Education of Ksar Said, University of La Manouba, Tunis 2010, Tunisia; 5Human Anatomy and Sport Medicine Division, Department of Public Health, University of Naples Federico II, 80131 Naples, Italy; stefanopalermi8@gmail.com; 6Sport Science Lab, 09131 Cagliari, Italy; ciao@migliaccio.it; 7Department of Philosophy, Sociology, Education, and Applied Psychology, Section of Applied Psychology, University of Padova, 35131 Padova, Italy; stefania.mannarini@unipd.it (S.M.); a.rossi@unipd.it (A.A.R.); 8Interdepartmental Center for Family Research, University of Padova, 35131 Padova, Italy

**Keywords:** field dance test, motor learning, physiological effort, sensitivity, training experience

## Abstract

The study aimed to explore the sensitivity and specificity of a new methodological approach related to the musical rhythm for discriminating a competitive Cuban dancer’s (CDCs) level. Thirty CDCs (Age 23.87 ± 1.76 years, body mass 60.33 ± 9.45 kg, stature 1.68 ± 0.07 m) were divided into three groups: beginner (BEG, *n* = 10), intermediate (INT, *n* = 10), and advanced (ADV, *n* = 10) according to their training experience/level. Each dancer was assessed while dancing at three different musical rhythms: fast (118 BPM), medium (96 BPM), and slow (82 BPM). The assessed variables were average heart rate (HR_M_), peak (HR_P_), and dancing time (DC_T_). The ADV group succeeded at all three musical combinations (317, 302, 309 s for 82, 96, 118 BPM). The INT group correctly performed only the first two combinations (304, 304 s for 82, 96 BPM), while a significant time difference was shown at the fast musical rhythm (198 ± 6.64 s) compared to the medium (*p* < 0.001) and slow rhythms (*p* < 0.001) respectively. As the speed of the musical rhythms increased, the BEG group was not able to follow the rhythm: their results were 300 ± 1.25 s for the slow musical rhythm, 94.90 ± 12.80 s for the medium musical rhythm and 34.10 ± 5.17 s for the fast musical rhythm (*p* < 0.001). The HR_M_ and HR_P_ grew along with the increase in musical rhythm for all groups (*p* < 0.001). The ROC analysis showed a high sensitivity and specificity in discriminating the groups for each rhythm’s condition. The BEG and INT groups showed an AUC = 0.864 (95% CI = 0.864–0.954); INT and ADV showed an AUC = 0.864 (95% CI = 0.864–0.952); BEG and ADV showed an AUC = 0.998 (95% CI = 0.993–1.000). The results of this study provided evidence to support the construct and ecological validity of the time of the musical rhythms related to competitive CDCs. Furthermore, the differences in the performances according to various musical rhythms, fast (118 BPM), medium (96 BPM), and slow (82 BPM), succeeded in discriminating a dancer’s level. Coaches and strength and conditioning professionals should include the Cuban Dance Field Test (CDFT) in their test battery when dealing with talent detection, selection, and development.

## 1. Introduction

Dance (DC) has been defined as an artistic human movement, which is formalized with qualities such as grace, elegance, and beauty, to the accompaniment of music or other rhythmic sounds and with the aid of mime, costumes, scenery, and lighting [1]. Dancing aims to communicate or express human emotions, themes, or ideas [1] and has been reported to involve a number of positive aspects [2]: it improves flexibility, postural stability, balance, physical reaction time, and cognitive performance [3], it increases the activity of the premotor cortex and impacts brain plasticity [4], it can improve balance [5] and strength [6] and reduce the risk of falls, and can prevent cognitive impairment [6].

Currently, the Latin American DC attracts huge public interest worldwide due to its expressive nature and is considered a social and cultural activity [7] for its potential as a community-based health-enhancing physical activity for adults [8]. Latin DC has been shown to play a suitable role in the engagement of individuals in physically active pursuits that are not necessarily thought of as traditional exercise per se [9]. The different styles of Latin DC (the construct validity) are related to the timing of steps, the movements on the DC floor and the dancer’s preferences for turns and moves, as well as attitude, dress code and other aspects. Despite its popularity, only a limited amount of research as explored the effects of DC on health promotion [10,11,12] or the physiological responses to DC exercise [13,14,15].

In recent decades, Cuban music, and Cuban dances have been exported all over the world, thanks to the great variety of DCs and rhythms. Currently, competitive Cuban dance style is in the process of becoming very popular in many countries across Europe, Latin America, and North America. Competitive Cuban dancers train rigorously to achieve a high level of performance to compete in international meetings [16]. This has allowed DC professionals to open DC schools by diversifying teaching classes according to the training experience of students and distributing classes into basic, intermediate, and expert levels: these classes provide three different regimens related to the musical intensity (slow, medium, and fast). 

Di Blasio et al. [17] found that Caribbean DC fits with the international guidelines for the maintenance of health. The authors [17] reported that the intensity of Caribbean DC is moderate and differs between beginner and advanced dancers, ranging from 3.42 to 3.81 MET, respectively. A more recent study [16] that assessed the energy expenditure and the intensity of physical activity during the various phases of the competitive simulation of Latin American DC demonstrated that competitive Latin American DC is a heavy exercise. The authors [16] suggested that monitoring variables during normal training can improve training protocols and the dancer’s fitness levels. Knowing that, the Cuban dance is performed at different dancer’s levels and with different rhythms, it could be interesting to set up an easy and practical field test to distinguish the dancer’s level for testing and training. Therefore, the aim of the present study was to explore the sensitivity and specificity of a new DC field test related to the rhythm synchronization to discriminate Cuban dancer’s level. It was hypothesized that this study will provide evidence to support the construct and ecological validity of the time musical rhythms related to competitive Cuban Dancers and that the 118-96-82 BPM test which will classify Competitive Cuban dancers as beginner, intermediate, or advanced can be included in the test battery when dealing with talent detection, selection, and training development.

## 2. Materials and Methods

The sample size was planned a priori using previous studies-with similar measures for outcomes and statistical analyses-which reported having enrolled a sample of up to 10 subjects per group [13,18]. Thirty Cuban dancers (age 23.87 ± 1.68 years, body mass 60.33 ± 9.45 kg, stature 1.67 ± 0.08 m, BMI 21.44 ± 1.82 kg·m^−2^) voluntarily participated in this study. The subjects (Table 1) were divided into three groups, according to their dance training experience and national ranking: beginner (BEG, *n* = 10: age 23.70 ± 1.70 years, body mass 61.40 ± 10.68 kg, stature 1.67 ± 0.09 m, BMI 21.77 ± 1.60 kg·m^−2^), intermediate (INT, *n* = 10: age 24.10 ± 1.54 years, body mass 59.20 ± 8.02 kg, stature 1.66 ± 0.07 m, BMI 21.36 ± 1.65 kg·m^−2^) and advanced (ADV, *n* = 10: age 23.80 ± 1.81 years, body mass 60.40 ± 9.66 kg, stature 1.68 ± 0.06 m, BMI 21.19 ± 2.17). Inclusion criteria for participation in this study were: a minimum training frequency of two sessions per week or more and being free from any injury or pain that would have prevented maximal effort during the manipulations. The ranking inclusion was final or semifinal participation in national Cuban Dance Competition for ADV in the last year, qualification for regional Cuban Dance Competition for INT in the last year, and qualification for city Cuban Dance Competition for BEG in the same year. All subjects gave their written informed consent to participate in the study after receiving a thorough explanation of the study’s protocol. The protocol conformed to internationally accepted policy statements regarding the use of human participants in accordance with the Declaration of Helsinki and was approved by the Institutional Review Board of Ovidius University of Constanta (851/2018) (date of approval: 11 October 2018).

### 2.1. Experimental Setting

A cohort study design was used. Three different musical rhythms [19] were chosen according to the Cuban dance related to the beats per minute (BPM): fast (118 BPM), medium (96 BPM) and low (82 BPM). The investigation took place in November 2018, when all sessions were conducted in a dance gym with similar environmental conditions (temperature and relative humidity for each session ranging from 22–24 °C and 2–27%, respectively). The rate of BPM was assessed with a BPM analyzer V1.0 (MixMeister Technology, LLC, Kirkland, WA, USA). The dancers were randomly assigned to the three different musical experimental conditions (fast-medium-low randomly, Figure 1) and assessed on three different days intersected by 24 h to avoid any influence of the fatigue, so each group concluded all three different musical rhythms. During each testing session, each subject performed a warm-up (10 min) that consisted of light running followed by a stretching sequence (5 min) [20]. The following variables were measured: heart rate (mean and peak, HR_M_ and HR_P_, respectively) and dance-time (DC_T_). Each dancer was given five minutes of rest between each test. At the end of the test, 30 s of low intensity stretching was performed for each muscle group to strengthen agonist muscles and the decrease the resistance of antagonist muscles [21]. During the testing sessions, each dancer wore Polar OH1 arm bands (Polar Electro, Kempele, Finland) on their upper right arm [22] to assess the heart rate. At the end of each dance, HR_M_ and HR_P_ (beats·min^−1^) and DC_T_ (s) were recorded by two experts as sport scientist and national CDC teacher. The test duration (time in s) was assessed by manual timing (using a simple stopwatch) accompanied by a video recording (40 frame·s^−1^). Video sequences were treated with VirtualDub software (V1.10.4) to verify the validity of manually timing of the execution time (ET). ET was defined as the time between the start signal (immediately after the first BPM music) and the noise of the slap at the second error on the musical rhythm. The exercise ended when mistake in the rhythm or DC steps were made twice in a row, according to the World Dance Sport Federation rules [23]: whenever the dancers were out of time, or made mistakes (style or technique) in the DC steps, the music was stopped, thus allowing the counting the errors and defining the DC_T_.

### 2.2. Statistical Analysis

Statistical analyses were performed with R statistical software [24]. Mean comparisons (ANOVA) were respectively performed to examine differences between the scores for dependent variables (HR_M_, HR_P_, and DC_T_) obtained for each trial. Considering the study design, a repeated-measure analysis of variance (ANOVA) with a grouping factor (within-between ANOVA; WB-ANOVA) was performed. The means of each variable was compared within the 3 different musical rhythms (fast as 118 BPM; medium as 96 BPM; and low as 82 BPM) between the three groups (beginner (BEG), intermediate (INT), and advanced (ADV)). Focused contrasts were also performed, assessing specific differences across each condition within and between each time. The strength of the differences was interpreted using the partial eta-squared ($\eta_{p}^{2}$) and its benchmarks: null ($\eta_{p}^{2}$ < 0.010), small ($\eta_{p}^{2}$ from 0.011 to 0.059), moderate ($\eta_{p}^{2}$ from 0.060 to 0.139), and large ($\eta_{p}^{2}$ > 0.140) [25]. Receiver Operating Characteristics (ROC) curves methodology—Youden estimation [26]—was used to assess the accuracy of discriminating each individual group with a fixed category time (s) for each musical rhythm with the confidence interval (CI). The global accuracy-validity was estimated with area under the ROC curve (AUC) with 5000 stratified bootstrap resamples—interpreted using the Sweets benchmarks [27,28]. The sensibility (Se), specificity (Sp), positive predictive value (PPV), negative predictive value (NPV), and accuracy (ACC) and precision (PRC) were also computed for each cut-off point. The significance level was set at priori with *p* < 0.05.

## 3. Results

### 3.1. Repeated Measure ANOVA between Groups

Considering the HR_M_ (Table 1 and Table 2), the within-between interaction was statistically significant (*F* = 11.831, *p* < 0.001; $\eta_{p}^{2}$ = 0.467) as well as the within-variable effect (*F* = 530.412, *p* < 0.001; $\eta_{p}^{2}$ = 0.952). Conversely, the between-variable effect was not significant (*F* = 2.162, *p* = 0.135; $\eta_{p}^{2}$ = 0.138). In terms of the HR_P_, the within-between interaction was statistically significant (*F* = 8.090, *p* < 0.001; $\eta_{p}^{2}\;$= 0.375) as well as the within-variable effect (*F* = 328.898, *p* < 0.001; $\eta_{p}^{2}$ = 0.924). In contrast, the between-variable effect was not significant (*F* = 0.132, *p* = 0.877; $\eta_{p}^{2}$ = 0.010). Considering the DC_T_, the within-between interaction was statistically significant (*F* = 1395.322, *p* < 0.001; $\eta_{p}^{2}$ = 0.990) as was the within-variable effect (*F* = 2976.946, *p* < 0.001; $\eta_{p}^{2}$ = 0.991), and the between-variable effect (*F* = 8625.259, *p* < 0.001; $\eta_{p}^{2}$ = 0.998). Focused contrasts were performed separately between the three performances in each group and between the three groups in each performance (Table 3 and Table 4).

### 3.2. Receiver Operating Characteristics Curves 

The collected data showed that ADV group completed all three musical rhythm combinations without errors (317, 302, 309 s for 82, 96, and 118 BPM, respectively), whereas the INT group only correctly performed the first two combinations (301.60, 303.90 s for 82, and 96 BPM, respectively), while at 118 BPM there was a significantly difference (198 ± 6.64 s, with *p* < 0.05 compared to the others). As the musical rhythms increased, the BEG group could not dance according to the rhythm: their results were 300 ± 25 s for 82 BPM, 94.90 ± 2.75 s for 96 BPM and 34.10 ± 5.17 s for 118 BPM. Considering these results, the ROC analysis confirmed the goodness of the proposed discrimination parameters. Indeed, the ROC curved showed a high sensitivity and specificity in discriminating each individual group (Figure 2, Appendix A) with a fixed category time (s) for each musical rhythm. More specifically, BEG and INT showed an AUC = 0.864 (95% CI = 0.864–0.954) with a cutoff value of 104.5 with a Se of 1 (95% CI: 1–1) and a Sp of 0.667 (95% CI: 0.500–0.833); true positives = 30; true negatives = 20; false positives = 10; false negatives = 0; PRC = 0.75; and ACC = 0.833. Moreover, INT and ADV showed an AUC = 0.864 (95% CI = 0.864–0.952) with a cutoff value of 305.5 with a Se of 7 (95% CI: 0.533–0.867) and a Sp of 0.900 (95% CI: 0.800–1) true positives = 21; true negatives = 30; false positives = 3; false negatives = 9; PRC = 0.875; and ACC = 0.800. Lastly, BEG and ADV showed an AUC = 0.998 (95% CI = 0.993–1.000) with a cutoff value of 300.5 with a Se of 1 (95% CI: 1–1) and a Sp of 0.933 (95% CI: 0.833–1) true positives = 30; true negatives = 28; false positives = 20; false negatives = 0; PRC = 0.937; and ACC = 0.967.

## 4. Discussion

This study was the first to assess the ecological validity of a field test designed to evaluate the Competitive Cuban dancer’s (CDC’s) level. The results indicated that with the increasing of BPM, the BEG group lose the ability to DC until the end of the musical sequence, compared to the INT and ADV groups (Table 3). It is clear how fundamental an examiner’s experience has been evaluating dancers’ level. So, our first hypothesis was confirmed. The test construct and ecological validity are necessary when no gold standards are available for the ability of interest (i.e., criterion validity) [29]. Moreover, ecological validity constitutes a valuable prerequisite for the test specificity and to promote the practical interest of a test [29] in this field.

The results showed significant differences between the groups that are important for the physically active population that participate in the world of DC and can be used to help improving training activity: indeed, a one-to-one personal workout is an effective method for changing attitudes and thereby increasing the amount of physical activity [30].

The ROC analysis showed that the choice of musical BPM can be useful in discriminating the dancer’s level. Thanks to the test carried out, it was clear that the time spent learning a good musical rhythm is a fundamental element in the ability to DC until the end of the music without making mistakes (Table 3). Furthermore, DC_T_ showed a strong interaction with musical rhythms (*p* < 0.001) and expertise (*p* < 0.001) between the groups (Table 2).

To the best of our knowledge, no previous study has ever done it. It is likely the BEG group, would have performed better after a greater number of training hours. This is in line with studies in other sporting activities: the more you train, the more you improve [31]. In this study, subjects performed three tests while dancing at three different musical rhythms. To date, little is known about the energy expenditure (EEs) of DC’s during Latin American DC competitions. Only one previous study has focused on measuring EEs during Latin American competitive DC, and this study examined the relationship between heart rates and oxygen consumption during submaximal exercise [32]. This lack of data may be due to the difficulties in applying metabolic meter devices during performances. Nevertheless, assessing EE is important for several reasons. First, EE data would provide a better understanding of the competition’s energy demands on professional dancers. This information would enable more specific training methods to be implemented to ensure that professional dancers have adequate fitness levels. Second, knowledge of the total EE and the intensity of DC activities could be helpful for individuals who enjoy using this social activity to improve their general health status [16]. The results indicated an increased HR_M_ and HR_P_ when the musical rhythms increased (82, 96, 118 BPM, Table 3). These findings show that competitive Latin American DC is a vigorous activity: the HR_M_ monitored at 118 BPM was 165.50 for the BEG, 175.50 for the INT and 175.70 for the ADV groups in agreement with previous reports [16]. Our results indicated that a field test based on time and rhythm can discriminate the level of the dancer DC’s fitness [33], and such assessments are frequently used in sport (i.e., Cooper test [34]).

Furthermore, our results could provide a practical method for a longitudinal study with a larger cohort of participants, since our samples was not wide. It could also be interesting to assess its validity for different styles of music and different rates of BPM since these aspects were limited in the present research. Another important application could be to use this method to set up a workout. If, following a correct training periodization, the results show significant differences, then the test could be said to have provided a very important contribution to the dance world, which could use the discovery to streamline and improve the learning period of the dancers’ musical rhythm.

Our test was designed to assess only one aspect of Cuban Competitive dance. Due to the lack of the application of training theories to the world of dance, more studies such as this one are necessary for the correct definition of training zones and the s of loads. While differences in the performance to musical rhythms succeeded in discriminating a dancer’s level, other important aspects such as the style and the level of artistic execution of the dance should also be taken into consideration when dealing with talent detection, selection, and development. The ability for a dancer to maintain a correct musical rhythm without technical errors does not include the ability to feel the music, and to communicate its emotion. Future studies should implement more tests while considering both the speed and the beauty of the exercise (the artistic aspect) of the Competitive Cuban Dance.

## 5. Conclusions

The study arises from the need to demonstrate with objective data what is already known but is not present in any literature at a scientific level: “one thing is to know it; another is to demonstrate it”. Specifically, the ability to maintain a correct musical rhythm for longer in the absence of errors is a prerogative of ADV dancers compared to INT and BEG dancers. The results of this study provided evidence to support the construct and ecological validity of the time musical rhythms related to CDC’s. Furthermore, differences in 118-96-82 BPM performance indicate a way to discriminate dancer’s level. Coaches and strength and conditioning professionals should include the Cuban Dance Field Test (CDFT) in their test battery when dealing with talent detection, selection, and training development.

## Figures and Tables

**Figure 1 ijerph-18-13287-f001:**
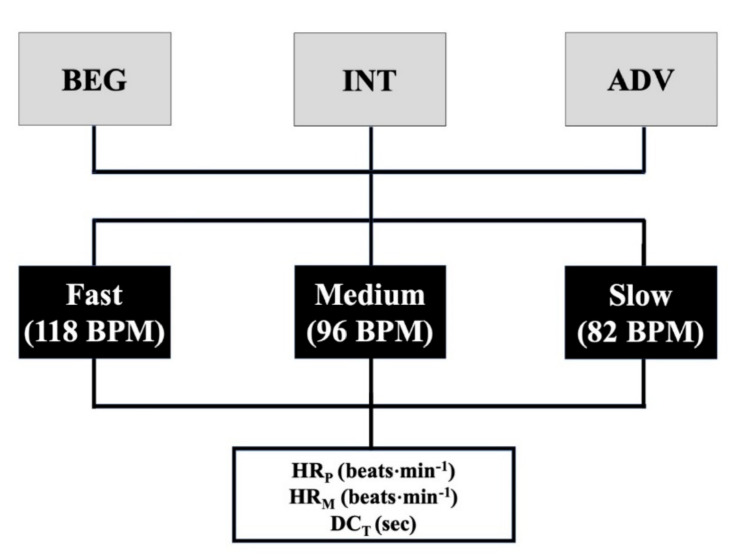
Schematic representation of the experimental design.

**Figure 2 ijerph-18-13287-f002:**
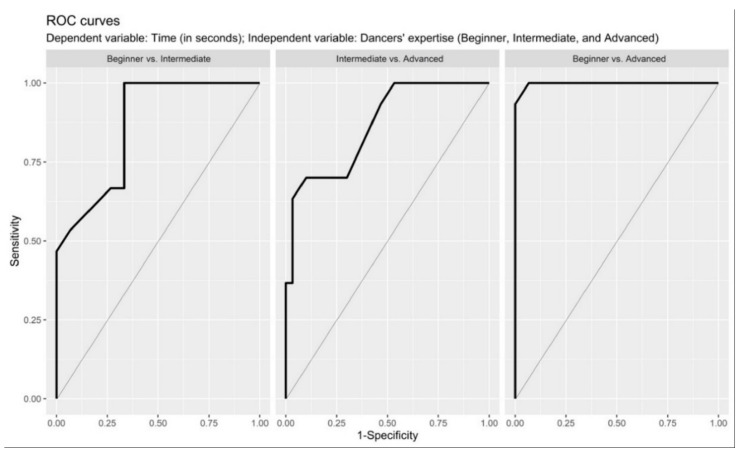
ROC in-between groups on performance time related.

**Table 1 ijerph-18-13287-t001:** Anthropometric, training experience and metabolic variables for each group.

Variables	Overall (*n* = 30)		Beginner (*n* = 10)		Intermediate (*n* = 10)		Advanced (*n* = 10)		*F*	*p*-Value
Age, y	23.87	1.677	23.70	1.705	24.10	1.539	23.80	1.808	0.456	*p* = 0.635
Body mass, kg	60.33	9.452	61.40	10.679	59.20	8.019	60.40	9.662	0.402	*p* = 0.670
Stature, m	1.68	0.076	1.67	0.088	1.6620	0.0774	1.6840	0.062	0.617	*p =* 0.542
BMI, kg·m^−2^	21.44	1.823	21.77	1.595	21.363	1.654	21.195	2.176	0.769	*p* = 0.455
Training Experience, y	4.750	4.354	0.250	0.000	4.100	0.738	9.900	2.486	81.894	*p* < 0.0001
Weekly training, h	13.033	9.152	3.000	0.000	11.800	1.476	24.300	3.622	224.634	*p* < 0.0001
HR_M_, beats·min^−1^	148.26	20.525	146.47	16.060	148.33	20.620	149.97	24.564	0.215	*p* = 0.807
HR_P_, beats·min^−1^	157.87	19.195	157.50	16.038	158.43	18.796	157.67	22.796	0.020	*p* = 0.980
DC_T_, s	240.22	101.453	142.90	115.817	268.00	50.155	309.77	7.686	42.443	*p* < 0.001

Note: mean, standard deviation, and one-way ANOVA for all variables studied.

**Table 2 ijerph-18-13287-t002:** General effect of musical rhythms (within variable), group expertise (between variable), and their interaction on HR_M_, HR_P_, and DC_T_.

Variables	*F*	*p*-Value	$\eta_{p}^{2}$
HR_M_ (beats·min^−1^)			
Rhythms (W)	530.412	*p* < 0.001	0.952
Expertise (B)	2.162	*p* = 0.135	0.138
Rhythms × Expertise (W × B)	11.831	*p* < 0.001	0.467
HR_P_ (beats·min^−1^)			
Rhythms (W)	328.898	*p* < 0.001	0.924
Expertise (B)	0.132	*p* = 0.877	0.010
Rhythms × Expertise (W × B)	8.090	*p* < 0.001	0.375
DC_T_ (s)			
Rhythms (W)	2976.946	*p* < 0.001	0.991
Expertise (B)	8625.671	*p* < 0.001	0.998
Rhythms × Expertise (W × B)	1395.322	*p* < 0.001	0.990

Note: For both “HR_M_” and “HR_P_”, sphericity assumption was violated and the Huynh-Feldt correction was applied.

**Table 3 ijerph-18-13287-t003:** Mean (standard deviation) for each variable and comparison across different musical rhythms in each group separately.

**Variables**	**Beginner**					Intermediate					Advanced				
	82 BPM	96 BPM	118 BPM	*F*	$\eta_{p}^{2}$	82 BPM	96 BPM	118 BPM	*F*	$\eta_{p}^{2}$	82 BPM	96 BPM	118 BPM	*F*	$\eta_{p}^{2}$
HR_M_ (beats·min^−1^)	127.9 (3.51)	145.9 (3.75)	165.6 (3.81)	271.88 **	0.968	127.5 (1.27)	142.0 (3.53)	175.5 (3.03)	965.44 **	0.991	120.8 (5.33)	153.4 (14.7)	175.7 (2.45)	103.04 **	0.920
HR_P_ (beats·min^−1^)	138.0 (2.05)	163.3 (12.3)	171.2 (2.44)	60.28 **	0.870	137.8 (1.23)	155.9 (7.47)	181.6 (2.12)	296.38 **	0.971	130.9 (6.37)	159.8 (12.4)	182.3 (2.11)	111.14 **	0.925
DC_T_ (s)	299.7 (1.25)	94.9 (12.8)	34.1 (5.17)	2698.03 **	0.997	301.6 (1.35)	303.9 (2.51)	198.5 (6.64)	2225.34 **	0.996	317.1 (6.33)	302.7 (2.06)	309.5 (5.43)	16.33 **	0.645

Note: ** *p* < 0.001.

**Table 4 ijerph-18-13287-t004:** Mean (standard deviation) for each variable and comparison between each group in different musical rhythms separately.

**Variables**	**82 BPM**					96 BPM					118 BPM				
	BEG	INT	ADV	*F*	$\eta_{p}^{2}$	BEG	INT	ADV	*F*	$\eta_{p}^{2}$	BEG	INT	ADV	*F*	$\eta_{p}^{2}$
HR_M_ (beats·min^−1^)	127.9 (3.51)	127.5 (1.27)	120.8 (5.329)	11.27 **	0.455	145.9 (3.75)	142.0 (3.53)	153.4 (14.7)	4.155 *	0.235	165.6 (3.81)	175.5 (3.03)	175.7 (2.45)	33.72 **	0.714
HR_P_ (beats·min^−1^)	138.0 (2.05)	137.8 (1.23)	130.9 (6.37)	10.59 **	0.440	163.3 (12.3)	155.9 (7.47)	159.8 (12.4)	1.141 ^§^	0.078	171.2 (2.44)	181.6 (2.12)	182.30 (2.11)	77.81 **	0.852
DC_T_ (s)	2999.7 (1.25)	301.6 (1.35)	317.1 (6.33)	62.85 **	0.823	94.9 (12.8)	303.9 (2.51)	302.7 (2.06)	2505.83 **	0.995	34.1 (5.17)	198.5 (6.64)	309.5 (5.54)	5672.07 **	0.998

Note: ^§^ *p* > 0.050 ns; * *p* < 0.050; ** *p* < 0.001.

## Data Availability

The data that support the findings of this study are available from the corresponding author, upon reasonable request and in the Appendix A.

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
