# Peer review of "Ecological and Construct Validity of a New Technical Level Cuban Dance Field Test"

_ijerph, 2021, doi:10.3390/ijerph182413287_

Round 1

Reviewer 1 Report

ijerph-1466702-peer-review-v1

Journal: International Journal of Environmental Research and Public Health

Ecological and Construct Validity of the Cuban Dance Technical Level Field Test

Thank you for the opportunity to review your manuscript. The topic is novel and interesting and seeks to define an ecologically valid of discriminating level of dance ability based on rhythm synchronisation in Cuban dance. The overall finding appeared to be that timed trials using three different music tempos successfully differentiated level of ability. 

Specific comments:

I believe that the manuscript could be greatly improved and made more readable with more careful editing of the English grammar, being a little more concise in sections, and limiting the scope to the study design and supporting literature. 

  • At times incorrect/confusing words or terms are used (e.g. L72  heavy exercise vs high intensity exercise), some sentences were difficult to follow. 
  • Lines 190 to 209 appeared to be well beyond the scope of the study

While the analysis of data is explained in detail I was not convinced of the usefulness of Tables 3 and 4 which don't appear to add much value. Your 2 way ANOVAs with repeated measures presented the key significant differences.

Overall, my view is this manuscript could be more readable and of interest to readers if you made some hard decisions on significantly reducing the the amount of data that you present.

You mention test specificity and sensitivity but don't real embrace these concepts in any detail. You have attempted to address this statistically with your ROC analysis but do not discuss the ability of your tests to successfully identify a participants level without false positives or false negatives. I would expect there would be clear cut points for these.

Your comments on strength and conditioning professionals and their role were unclear and I wondered whether the relationship between physical fitness elements, skill and ability to read and maintain tempo (?motor control abilities) were discussed significantly. The ANOVAs show no interaction between expertise and rhythm for either HR mean or HR peak?

Specific questions/comments

  1. You don’t provide detail on the 2 key criteria for your dancer groupings - dance training experience and national rankings - this is more important for readers than age and anthropometrics.
  2. Bpm became confusing as you appear to use it interchangeable for music tempo and heart rate.
  3. I think you mean peak HR not maximum HR.

Author Response

1st reviewer

ijerph-1466702-peer-review-v1

Journal: International Journal of Environmental Research and Public Health

Ecological and Construct Validity of the Cuban Dance Technical Level Field Test

Thank you for the opportunity to review your manuscript. The topic is novel and interesting and seeks to define an ecologically valid of discriminating level of dance ability based on rhythm synchronization in Cuban dance. The overall finding appeared to be that timed trials using three different music tempos successfully differentiated level of ability. 

Author’s response: The authors wish to thank the reviewer for their valuable comments. These have helped improve the paper.

Specific comments:

I believe that the manuscript could be greatly improved and made more readable with more careful editing of the English grammar, being a little more concise in sections, and limiting the scope to the study design and supporting literature. 

Author’s response: We thank the reviewer for these suggestions. The authors worked to improve the main document according flowing to the English proofreading service.

-We moved the following sentences from the discussion to the introduction section to limit the scope to the study: “ Dance have been reported to involve a number of positive aspects [26]: it improves flexibility, postural stability, balance, physical reaction time, and cognitive performance [28]; it increases the activity of the premotor cortex and impact brain plasticity [30]; it can improve balance [31] and strength [32]; and reduce the risk of falls, and prevent from cognitive impairment [32].”

-We also erased the following sentence from the introduction section to be more concise in sections: “Wininger and Pargman [2] indicated that satisfaction with the music is the best predictor of exercise enjoyment.”

-We also erased the following sentence from the discussion section to be more concise in this section.

“The adoption of pleasant programs of leisure-time physical activity may be a reasonable option to increase motivation and adherence [32], while maintaining a supervised control on the amount of physical activity.”

“Among leisure time activities, dancing has a remarkable place [33] when arranged with patients’ associations, aerobic DC, Latin DC, Folk DC, and standard Ballroom DC and may all help aging people to enjoy their physical activity.”

-The acoustic stimulation and the music might strengthen the beneficial effects of aerobic exercise on cognitive functions [34]

- In order to be more concise in supporting literature, We removed 5 references from the manuscript: [27], [29],  [32], [33] , [34], and [2] related to the first submission.

At times incorrect/confusing words or terms are used (e.g. L72 heavy exercise vs high intensity exercise), some sentences were difficult to follow. 

Author’s response: We agree with this suggestion, therefore We used “heavy exercise” according to Massidda et al (https://pubmed.ncbi.nlm.nih.gov/22211197/)

Lines 190 to 209 appeared to be well beyond the scope of the study

Author’s response:  The authors deleted Lines 190 to 209 from the discussion section.

While the analysis of data is explained in detail, I was not convinced of the usefulness of Tables 3 and 4 which don't appear to add much value. Your 2-way ANOVAs with repeated measures presented the key significant differences.

Author’s response:  we thank the reviewer for the comment. However, we think that given the amount of data reported, Tables 3 and 4 may help the reader – even an inexperienced one – to read more fully the findings of our study. If this is not necessary, we would prefer not to remove them from the text.

Overall, my view is this manuscript could be more readable and of interest to readers if you made some hard decisions on significantly reducing the amount of data that you present.

Author’s response: We agree with this opinion and decision, but We included all data to better replicate this study and to compare it with other studies

You mention test specificity and sensitivity but don't real embrace these concepts in any detail. You have attempted to address this statistically with your ROC analysis but do not discuss the ability of your tests to successfully identify a participant’s level without false positives or false negatives. I would expect there would be clear cut points for these.

 Author’s response:  we are thankful to the reviewer for this comment. Indeed, in the last version of the manuscript we miss to insert values of sensitivity and specificity. Consequently, we add them to the dedicated section of the results. Moreover, we provide the complete values into a supplementary file.

Indeed, the ROC curved showed a high sensitivity and specificity to discriminate each individual group (Figure 2 and Supplementary file) with fixed category time (sec) for each musical rhythm. More in detail, BEG and INT showed an AUC = 0.864 (95% CI = 0.864 – 0.954) with a cutoff value of 104.5 with a Se of 1 (95%CI: 1 – 1) and a Sp of 0.667 (95%CI: 0.500 – 0.833); true positives = 30; true negatives = 20; false positives = 10; false negatives = 0; PRC = 0.75; and ACC = 0.833. Moreover, INT and ADV showed an AUC = 0.864 (95% CI = 0.864 – 0.952) with a cutoff value of 305.5 with a Se of 7 (95%CI: 0.533 – 0.867) and a Sp of 0.900 (95%CI: 0.800-1) true positives = 21; true negatives = 30; false positives = 3; false negatives = 9; PRC = 0.875; and ACC = 0.800. Lastly, BEG and ADV showed an AUC = 0.998 (95% CI = 0.993 – 1.000) with a cutoff value of 300.5 with a Se of 1 (95%CI: 1 – 1) and a Sp of 0.933 (95%CI: 0.833 – 1) true positives = 30; true negatives = 28; false positives = 20; false negatives = 0; PRC = 0.937; and ACC = 0.967.

Your comments on strength and conditioning professionals and their role were unclear and I wondered whether the relationship between physical fitness elements, skill and ability to read and maintain tempo (motor control abilities) were discussed significantly. The ANOVAs show no interaction between expertise and rhythm for either HR mean or HR peak?

 Author’s response: the interaction was found as showed in Table 3 (Rhythms * Expertise (W * B))

Specific questions/comments

  1. You don’t provide detail on the 2 key criteria for your dancer groupings - dance training experience and national rankings - this is more important for readers than age and anthropometrics.

Author’s response:  We included in Table 1 the training experience (y) and the weekly training (h)

  1. Bpm became confusing as you appear to use it interchangeable for music tempo and heart rate.

Author’s response:  We agree with this comment, Therefore, to better discriminate it we used BPM (for musical rhythm) and HRM and HRP (for mean and peak heart rate, respectively).

  1. I think you mean peak HR not maximum HR.

Author’s response: Yes, peak as “HR peak”, all main document was updated with peak HR peak (HRP)

Reviewer 2 Report

The study aimed to explore the sensitivity and specificity of a new methodological approach related to the rhythm synchronization to discriminate Competitive Cuban dancer’s level. The authors concluded that this study provided evidence to support the construct and ecological validity of the musical rhythms time related to competitive Cuban Dancers. Also, differences on 118-96-82 bpm performance explained to discriminate dancer’s level. From a practical application of view, coaches and strength and conditioning professionals should include the 118-96-82 bpm in their test battery when dealing with talent detection, selection, and development.

The manuscript is well written and present interesting findings. However, some modifications are required:

I suggest presenting a clear hypothesis at the end of the Introduction and after the aim of the study. This will help readers.

Do the authors calculate the required sample size before the study?

I suggest adding a schematic representation of the experimental design.

In the statistical analysis, I suggest adding the p level and the confidence interval.

Table 1. Add the units for the measurement for all variables.

Please add a discussion of the limitations of the study.

Author Response

2nd reviewer

The study aimed to explore the sensitivity and specificity of a new methodological approach related to the rhythm synchronization to discriminate Competitive Cuban dancer’s level. The authors concluded that this study provided evidence to support the construct and ecological validity of the musical rhythms time related to competitive Cuban Dancers. Also, differences on 118-96-82 bpm performance explained to discriminate dancer’s level. From a practical application of view, coaches and strength and conditioning professionals should include the 118-96-82 bpm in their test battery when dealing with talent detection, selection, and development.

Authors’ Response (AR): The authors wish to thank the reviewer for their valuable comments. These have helped improve the paper.

The manuscript is well written and present interesting findings. However, some modifications are required:

I suggest presenting a clear hypothesis at the end of the Introduction and after the aim of the study. This will help readers.

Authors’ Response (AR): We thank the reviewer for this mention. the authors added in introduction subsection (line 78): “It was hypothesized that this study will provide evidence to support the construct and ecological validity of the musical rhythms time related to competitive Cuban Dancers and that the 118-96-82 BPM test will classify Competitive Cuban dancers as beginner, intermediate, and advanced in order to be included in the test battery when dealing with talent detection, selection, and development”.

Do the authors calculate the required sample size before the study?

Author’s response:  we thank the reviewer for the comment. Yes, the sample size was planned before data collection: The required sample size was based on previous scientific literature e.g.: (Guidetti et al. 2000; Blanksby and Reidy, 1998). To highlight this point, we add a sentence at the beginning of ‘Participants’ section. The sample size was planned a priori using previous studies – with similar outcomes measures and statistical analyses – which reported having enrolled a sample of up to 10 subjects per group (Guidetti et al. 2000; Blanksby and Reidy, 1998)

I suggest adding a schematic representation of the experimental design.

Authors’ Response (AR): We added a new Figure with a schematic representation of the experimental design.

In the statistical analysis, I suggest adding the p level and the confidence interval.

Author’s response: We updated the statistical analysis section with the p level and the confidence interval

Table 1.  Add the units for the measurement for all variables.

Authors’ Response (AR): We included all unit of measure.

Please add a discussion of the limitations of the study.

Authors’ Response (AR): The authors added some sentences dealing with the study limitations to the discussion subsection (from line 241 to 251): “Our test was designed to assess only one aspect (the speed) of Competitive CDC. Due to the lack of application of training theories in the world of DC, more studies in this topic are necessary for a correct definition of training zones and the specification of loads. While differences on BPM performance succeeded to discriminate DC’s level, other important aspects such as the level of the artistic execution (grace, elegance, and beauty) of the DC should also be taken into consideration when dealing with talent detection, selection, and development. The ability for a dancer to maintain a correct musical rhythm without technical errors does not include his ability to feel the music, and to communicate the emotion. Future studies should implement more tests while taking into account both the speed and the artistic aspect of the exercise of the Competitive CDC.”

Reviewer 3 Report

The aim of the present study was to explore the sensitivity and specificity of a new dance field test related to the rhythm synchronization to discriminate Cuban dancer’s level.

Although the article is not very novel, it is well presented and proposes a practical method for the classification of dancers.

There is a lack of application of training theories to the world of dance. More studies like this one are necessary for the correct definition of training zones and the specification of loads.

Here are some contributions:

How you dance is not taken into account, only the speed of the dance is valued. Isn't it true that in the competitions the style is also valued? Can this method classify with precision without taking into account how they dance?

It is recommended to use abbreviations for words that are often repeated in the text, e.g. physical education.

Are there really no limitations to this study?

Author Response

3rd reviewer

The aim of the present study was to explore the sensitivity and specificity of a new dance field test related to the rhythm synchronization to discriminate Cuban dancer’s level.

Although the article is not very novel, it is well presented and proposes a practical method for the classification of dancers.

Authors’ Response (AR):  To the best of the authors knowledge, This study was the first to propose a test that was able to discriminate the DC’s level taking in account the time of the dance. There is a lack of application of training theories in the world of DC. More studies like this one are necessary for a correct definition of training zones and the specification of loads.

How you dance is not taken into account, only the speed of the dance is valued.

Authors’ Response (AR):

The authors added some sentences to the discussion subsection (from line 244 to 251): “While differences on BPM performance succeeded to discriminate DC’s level, other important aspects such as the level of the artistic execution (grace, elegance, and beauty) of the DC should also be taken into consideration when dealing with talent detection, selection, and development. The ability for a dancer to maintain a correct musical rhythm without technical errors does not include his ability to feel the music, and to communicate the emotion. Future studies should implement more tests while taking into account both the speed and the artistic aspect of the exercise of the Competitive CDC.”

Isn't it true that in the competitions the style is also valued?

Author’s response:  We agree with the style, infact one Teacher was present for this investigation and each participant was stopped if the technique or style was altereted, furthermore the style and technique was altereted when each participant was not able to follow the musical rhythm

Can this method classify with precision without taking into account how they dance?

Authors’ Response (AR): The 118-96-82 BPM test differenciated successfully Competitive Cuban dancers as beginner, intermediate, and advanced. The dancers had to complete the trials without undertaking technical errors. This means that how they dance is also valued. However, we consider that the style and the artistic aspects of the Competitive Cuban Dance should be the focus of future research in this topic.

It is recommended to use abbreviations for words that are often repeated in the text, e.g. physical education.

Authors’ Response (AR): The authors added some abbreviations for words that are often repeated in the text: DC: Dance; CDC: Cuban Dance; DCT: Dance time.

Are there really no limitations to this study?

Authors’ Response (AR): The authors added some sentences dealing with the study limitations to the discussion subsection (from line 241 to 251): “Our test was designed to assess only one aspect (the speed) of Competitive CDC. Due to the lack of application of training theories in the world of DC, more studies in this topic are necessary for a correct definition of training zones and the specification of loads. While differences on BPM performance succeeded to discriminate DC’s level, other important aspects such as the level of the artistic execution (grace, elegance, and beauty) of the DC should also be taken into consideration when dealing with talent detection, selection, and development. The ability for a dancer to maintain a correct musical rhythm without technical errors does not include his ability to feel the music, and to communicate the emotion. Future studies should implement more tests while taking into account both the speed and the artistic aspect of the exercise of the Competitive CDC.”

Round 2

Reviewer 1 Report

Well done editing - it reads much better

From my perspective there is still too much irrelevant data - tables 3 and 4 could be covered with a couple of sentences.

Your comments on strength and conditioning professionals and their role were unclear and I wondered whether the relationship between physical fitness elements, skill and ability to read and maintain tempo (motor control abilities) were discussed significantly. The ANOVAs show no interaction between expertise and rhythm for either HR mean or HR peak?

Also I am still not clear what a strength and conditioning coach would do with this information?

Author’s response: the interaction was found as showed in Table 3 (Rhythms * Expertise (W * B)) This rhythm and expertise

You don’t provide detail on the 2 key criteria for your dancer groupings - dance training experience and national rankings - this is more important for readers than age and anthropometrics.
Author’s response: We included in Table 1 the training experience (y) and the weekly training (h)

yes but you state that 'divided into 3 three groups, according to their dance training experience and national ranking: - do years training equate to training experience? where does national ranking come into it? could a poor dancer who has trained for 10 years be included in the ADV group? In the absence of those explanations it is hard to discern the real sensitivity and specificity - i.e. I'm not sure how rigorous your ranking was and whether one of the researchers assigned groupings. 

Author Response

Reviewer 1

Well done editing - it reads much better

Authors response: thanks for your suggestions

From my perspective there is still too much irrelevant data - tables 3 and 4 could be covered with a couple of sentence

Authors response: We partially agree with your perspective, but We would like to show all data to better replicate this investigation, differently the Results section will be heavier and difficulty for the readers understand each detail

Your comments on strength and conditioning professionals and their role were unclear and I wondered whether the relationship between physical fitness elements, skill and ability to read and maintain tempo (motor control abilities) were discussed significantly. The ANOVAs show no interaction between expertise and rhythm for either HR mean or HR peak?

Author’s response: the interaction was found as showed in Table 3 (Rhythms * Expertise (W * B)) This rhythm and expertise

Authors response: We are sorry, there was a mistake about the Table 3 while correctly in Table 2 the interaction was found for HRM, HRP, DCT (Rhythms * Expertise (W * B) respectively).

Also I am still not clear what a strength and conditioning coach would do with this information?

Authors response: In conclusion section we included several sentences about the practical application, particularly “Furthermore, differences in 118-96-82 BPM performance indicate a way to discriminate dancer’s level”

You don’t provide detail on the 2 key criteria for your dancer groupings - dance training experience and national rankings - this is more important for readers than age and anthropometrics.
Author’s response: We included in Table 1 the training experience (y) and the weekly training (h)

yes but you state that 'divided into 3 three groups, according to their dance training experience and national ranking: - do years training equate to training experience? where does national ranking come into it? could a poor dancer who has trained for 10 years be included in the ADV group? In the absence of those explanations it is hard to discern the real sensitivity and specificity - i.e. I'm not sure how rigorous your ranking was and whether one of the researchers assigned groupings. 

Authors response: We agree with your specific comment, therefore We included the ranking level for each group in method section (Inclusion Criteria) The ranking inclusion was final or semifinal in national Cuban Dance Competition for ADV in the last year, qualification for regional Cuban Dance Competition for INT in the last year, and qualification for city Cuban Dance Competition for BEG in the same year

Reviewer 2 Report

I suggest that the manuscript is now suitable for publication.

Author Response

Reviewer: I suggest that the manuscript is now suitable for publication.

The authors are grateful for your suggestions useful to improve the main document